# Effect of Research Impact on Emerging Camel Husbandry, Welfare and Social-Related Awareness

**DOI:** 10.3390/ani10050780

**Published:** 2020-04-30

**Authors:** Carlos Iglesias Pastrana, Francisco Javier Navas González, Elena Ciani, Cecilio José Barba Capote, Juan Vicente Delgado Bermejo

**Affiliations:** 1Department of Genetics, Faculty of Veterinary Sciences, University of Córdoba, 14014 Córdoba, Spain; ciglesiaspastrana@gmail.com (C.I.P.); juanviagr218@hotmail.com (J.V.D.B.); 2Department of Biosciences, Biotechnologies and Biopharmaceutics, Faculty of Veterinary Sciences, University of Bari ‘Aldo Moro’, 70121 Bari, Italy; elena.ciani1976.ec@gmail.com; 3Department of Animal Production, Faculty of Veterinary Sciences, University of Córdoba, 14014 Córdoba, Spain; cjbarba@uco.es

**Keywords:** animal welfare, bibliometrics, camels, emerging industry, international research, science–society dialogue, law enforcement

## Abstract

**Simple Summary:**

Transactional agreements between applied research and regulatory agencies in animal welfare are scarce for minority species. In the present study, camel science upturn and its academic and societal impacts are bibliometrically traced across academic journals involving camel referring documents. The journal, author number, corresponding author origin, discipline and publication year may affect camel research outcomes. Despite camel-related research and its mean impact factor having noticeably increased over the past three decades due to growing social and economic interests in their breeding, parallel evolution of specific welfare laws is limited. Reliable guidance and mandatory standard policies for assessing reared-camel welfare research are identified as primary requirements within this emerging industry on a global scale. Research must play a pivotal role in the formulation of regulations, as the disconnection between science and law renders the efforts to ensure sustainable camel husbandry practices under the scope of welfare impractical.

**Abstract:**

The lack of applied scientific research on camels, despite them being recognized as production animals, compels the reorganization of emerging camel breeding systems with the aim of achieving successful camel welfare management strategies all over the world. Relevant and properly-framed research widely impacts dissemination of scientific contents and drives public willingness to enhance ethically acceptable conditions for domestic animals. Consumer perception of this livestock industry will improve and high-quality products will be obtained. This paper draws on bibliometric indicators as promoting factors for camel-related research advances, tracing historical scientific publications indexed in ScienceDirect directory from 1880–2019. Camel as a species did not affect Journal Citation Reports (JCR) impact (*p* > 0.05) despite the journal, author number, corresponding author origin, discipline and publication year affecting it (*p* < 0.001). Countries with traditionally well-established camel farming are also responsible for the papers with the highest academic impact. However, camel research advances may have only locally and partially influenced welfare related laws, so intentional harming acts and basic needs neglect may persist in these species. A sustainable camel industry requires those involved in camel research to influence business stakeholders and animal welfare advocacies by highlighting the benefits of camel wellbeing promotion, co-innovation partnership establishment and urgent enhancement of policy reform.

## 1. Introduction

Old World camels (*Camelus dromedarius* or one-humped camel, *Camelus bactrianus* or two-humped camel and the wild species *Camelus ferus*) are mainly found in the desert and semi-desert areas of the Middle East through northern India and arid regions in Africa [1]. Still, some feral populations of dromedary camels inhabit in some arid regions of central Australia, India and Kazakhstan [2], and could also be found in the southwestern United States until the early 20th century [3]. However, the critically endangered wild Bactrian (*C. ferus*) only survives in remote areas of northwest China and Mongolia [4].

The domestication process of these genus presumably started about 4000 years ago in southern Arabia [5], and the main purposes for which these animals were utilized were its meat, milk, wool, hair and dung, or for draft purposes [6,7]. Domestic camels played an important role in Old World ancient nomadic civilizations’ prosperity. A few low-income nomadic livelihoods in Africa and Eastern Asia still depend on these pluripotential animals [4]. Camels play a pivotal role as food providers, even under extreme environmental situations (severe drought periods or in rural livelihood areas where other domestic animals struggle to thrive) during which directly-dependent human nutrition is precarious [1].

Nonetheless, the camel may have been one of the most ignored species compared with other domestic livestock from a productive, political or socioeconomic perspective. This inconsideration may have its basis on their usual relation with under-developed areas, which led to the misattribution of a low economic value and the underestimation and disregard by science of the potential of these animals as a multifunctional resource for humans with very low maintenance requirements [8].

Fortunately, the international relevance of camels has progressively increased due to their recognition as a sustainable livestock species worldwide. Such a distinctive position is strongly linked to the current concern and need to provide functional solutions to environmental emergencies like global warming or desertification [5,9]. Besides, as nomadic pastoralists progressively sedentarizate, nomad livestock-breeding simultaneously turns into intensive socialist livestock-breeding [10]. Other deeply rooted traditional activities stemming from the Bedouin ancestral culture, such as camel racing, have also become a remarkable profitable interest in Middle East Arab countries [11].

As a result, these growing social and economic interests in camel husbandry experienced during the past three decades [5] have parallelly promoted an increase in the scientific actions that are implemented and which deal with almost any discipline applied to the species. Regardless of the existence of widely applied scientific knowledge about anatomy, physiology and pathology in camels, planned research and codes of action on the best handling practices ensuring a sufficient welfare status in these animals are scarce and shallow [12].

In this context, addressing the basic requirements and making efforts to seek niches of further development and reorientation of the camel industry becomes crucial. However, these objectives can only be achieved if they are understood and assimilated from people’s perspectives on animal welfare and considering the social, political and economic causes that may support the new structures in the long term [13]. When attempting to assess quality of life, Fraser [14] proposed three basic conceptual judgments of animal welfare: biological functioning, affective state and natural living. As domestic animals, camels share their natural living with humans. Intensive management could be affecting their natural behavioral repertoire, leading to the development of stereotypies [15]. Hence, owners and industry users need to be provided with the resources, skills and proficiency to handle them properly, to preserve their wellbeing and to improve their productivity in a sustainable manner.

This social awareness is grossly affected by science and vice versa, as the binomial science–society is immersed in a process of diachronic evolution with a sufficiently proven cultural base. Aiming to reach a consensus on camel wellbeing, science must revise the existing legislative rules and identify the changes needed to adapt them to the new challenges that this rather overlooked multipurpose species faces nowadays. Thus, research outputs may encourage lawmakers to develop official regulations and highlight the need for expanded awareness when aiming to meet new societal values and expectations [16].

Under this theoretical framework, the present paper primarily aimed to evaluate the evolution of research advances and their scientific impact in regards to the camel species, considering potential conditioning factors such as the journal, number of authors per contribution, country of corresponding author, topic with which the different publications dealt, year of publication and camel species studied. We elaborated a map to depict the countries and disciplines reporting the highest level of expertise and research approaches in camel science, respectively. Bibliometric mapping enables visualization of scientific developments leading to an active involvement of stakeholders in the different subfields. Secondly, the demonstrable contribution that camel international research progress has made within the legal sphere was evaluated by tracing the number and academic content of the regulations implemented in terms of camel welfare promotion.

This conglomerate of information is intended to become a substantial reference source for academics studying these species, on the preliminary process of decision-making for which discipline may require further research development, identifying potential research collaborators and finding journals in which to publish the outcomes of such research. Additionally, legislative authorities could also appeal to and officially fund research projects in unexplored areas in camel science. The scientific outcomes may be translated into mandatory codes of practices, intending to achieve humane husbandry and care for and across all types of camel business.

## 2. Materials and Methods

### 2.1. Study Premises: Data Obtention

The present methodology is based on a previous manuscript by McLean and Navas Gonzalez [17]. After typing the word ‘camel’ in the searching window of www.sciencedirect.com on 31 December 2019, we exported the results obtained into a .xlsx file. The reason why we chose to use www.sciencedirect.com is that the tool present in the website allows us to extract data for its analysis in a way that other platforms such as https://www.ncbi.nlm.nih.gov/pubmed/ do not. Therefore, the public filters that can be implemented are not exhaustive enough to perform the proper analysis required for this retrospective observational longitudinal study over the period of time between 1880 and 2019, both inclusive.

Data were filtered to discard those documents which did not focus on the *Camelus* genus or its species (*Camelus dromedarius*, *Camelus bactrianus*, *Camelus ferus* and other extinct species) by searching for the words ‘camel/s’, ‘camelid/s’, ‘*Camelus*’, ‘dromedary/ies’, ‘Bactrian’ and ‘feral’ in each article. Papers containing unrelated homophone or analogous terms were discarded due to the lack of connection to the species with which this manuscript deals.

The documents selected were included in a database which comprised individual registries for each article. Each individual record consisted of the name of the journal in which the article was published, the Journal Citation Report (JCR) impact factor of the journal in the year in which the document was published, the mean JCR impact factor per journal in the whole period (1880–2019), the total number of citations of each paper, the mean number of citations per journal, the number of authors contributing to each publication, the country of the corresponding author, the topic or research area, the year of publication, the camel species being studied and the publication’s doi in order to trace the documents back to the internet site in which they are available.

JCR impact factor was registered per journal and year by consulting the Journal Citation Reports on Web of Science. A value of zero was given in the database for all articles published in non-indexed journals in its year of publication (n = 254). The total number of citations per paper was assessed through the same scientific database. Table 1 reports a summary of the conditioning factors of paper impact related variables (Table 2) considered in the model used and the level within such factors.

Official regulations which specifically mention camels as production or companion animals and give advice and establish responsibility for persons or leaders in charge of them were traced back and evaluated in the online database of the Global Animal LawAssociation (https://www.globalanimallaw.org/database/index.html). It contains all the available and updated information until 1 January 2020 in the fields of law making, law enforcing, lobbying and scientific knowledge in animal protection from a local to universal level.

### 2.2. Preliminary Statistics Assumption Testing

The Shapiro–Francia normality test of the test and distribution graphics package of the Stata Version 15.0 software (StataCorp LLC, College Station, TX, USA was used to test the normality (Appendix A). The rest of the parametric assumptions (Levene’s test to evaluate homoscedasticity, Mauchly’s W test to evaluate sphericity and Tolerance and Variance Inflation Factor to test for multicollinearity, respectively) were performed using SPSS Statistics for Windows statistical software, Version 25.0. IBM Corp., Armonk, NY, USA).

### 2.3. Statistical Analysis

As preliminary tests and our study data had violated parametric assumptions, a nonparametric approach was suggested. A summary of the median and mode for the variables and predictors assessed in this study is reported in Table 2. 

Initially, to determine the general evolution of camel research and its impact, a Kruskal–Wallis H test, Dunn test and Bonferroni correction were performed to identify differences in the distribution of the Journal Citation Report impact factor per paper publication year, the mean JCR impact factor per journal in the whole period, the total number of citations of papers and the mean number of citations per journal across levels of the variables name of the journal, number of authors, country of corresponding author, topic, year of publication and camel species studied while reducing the likelihood for an increased Type I error. Type I errors could potentially derive from redundancies resulting from the inclusion of an excessive number of factors (considering the relatively limited sample of our study) that are reducing the possibility to falsely detect an effect (noise), which indeed is not present, as a result of the inclusion of multiple variables (noise variables).

Afterwards, as a way to evaluate the second aim, an additional Kruskal–Wallis H test, Dunn test and Bonferroni correction were performed to identify and describe differences in camel research impact-related variables across the periods running from the publication of an international regulation and the following one. According to this criterion, the periods determined were 1880–2006, 2007–2011, 2012–2014, 2015 and 2016–2019.

These statistical analyses were aimed at evaluating the association between camel research progress and camel-related laws by opposing the specific topics and the critical points that these regulations approached to the scientific advances occurring during the period before each legislation was released. In the case that a previous legislation existed, advances considered were those comprised by the period between the previous legislation and the new one. Topics and critical points were quantitatively and qualitatively determined after empirical comprehensive examination of the content of regulation sources. In that connection, the mandatory or voluntary character and geographical scope of the application of such legislations are contemplated as the absence of derived legal responsibility and regional endowment policies can cause these guides to be overlooked.

Kruskal–Wallis H is based on a single independent factor accounting for the variance explained of a dependent variable with no additional factor contributing to the explanation of such variance at the same time. If a factor has reported a significant effect, then all levels in the same factor must be evaluated in pairs until all possible combinations have been tested.

Out of all possible comparison pairs, only statistically significant pairwise comparisons were considered by the Dunn test. Once pairs between which significant difference existed had been identified, a test of independence of the median was performed to detect differences in the median for the variable of ‘Journal Impact Factor’ across levels of the same factor.

Then, categorical regression (CATREG) with the Optimal Scaling Procedure from the Regression task in SPSS Statistics for Windows, Version 25.0, IBM Corp. (2016) was applied to issue specific regression equations to predict how research impact (scored through the dependent variables of JCR impact factor per paper publication year, mean JCR impact factor per journal and year, total citation number per paper and mean total citation number per journal, Table 2) linearly depended on the predictors which nonparametric tests determined to be significant (*p* < 0.05).

CATREG analysis can be used to summarize linear relationships between dependent variables that are simultaneously influenced by a set of independent variables. R squared was used to determine the ability of the model comprising the independent variables or factors reported in Table 1 to capture the variability in the continuous variables describing impact factor. R squared has also been defined as the coefficient of determination of a certain model. In these regards, higher R squared values may be a sign of smaller differences between observed data and fitted values derived from the application of the model.

Contextually, when factors lack a certain unit of measure (such as ordinal or categorical ones) or the units for the factor comprised within a certain model differ, β standardized coefficients should be used to interpret and compare their effects on our dependent variables. This way, models using standardized coefficients can be compared as a result of the intercept in each model being reduced to 0.00 after the standardization process. Following the common notation models, the regression equations for each predictor variable were Y_n_ = β_n_Z_n_ + ε, where Y_n_ is the n variable predictor, β_n_ is the regression coefficient for the n variable obtained in the n main component, Z_n_ is the score obtained in the field for n variable and ε represents the estimation error. Specific regression equations are reported in the Regression Coefficients subsection of the Results section of the present manuscript.

During the process of evaluation of β standardized coefficients, 0.632 bootstrap cross-validation was used to estimate the prediction error of the CATREG model, provided our sample size was sufficient for the number of predictors comprised in the model in order for 0.632 bootstrap to be reliably computed as suggested by other authors [18].

As impact factor variables could somehow relate to the result of the methods used for them to be determined, to evaluate the correlation between the impact factor related variables compromised in our study, Spearman’s rank-order correlation was performed using the Bivariate task of the Correlate Procedure in SPSS Statistics for Windows, Version 25.0, IBM Corp. (2016) (Appendix A).

## 3. Results

### 3.1. Database Filtering Process: Study Sample

The word ‘camel’ was included in 24,611 results on first search in ScienceDirect site. From 2000 until 2019, the number of results reported was 13,932, while the rest (10,679) were published between 1880 and 1999. For this 139-year period (1880–2019), the mean number of articles per year was 177. The study sample comprised publications until 2019 as this was the last complete year when the study was performed.

When sorted depending on their type, total publications including the word ‘camel’ comprised 1520 reviews, 14,122 research articles, 597 encyclopedia entries and 3060 book chapters. Approximately 5312 articles, conference abstracts, case reports, data articles or short communications had been published under an open access policy.

Those publications which did not focus on the *Camelus* genus or its species (rather allude to the term, not the animal itself) were discarded. In the end, the study sample comprised 1011 articles (135 were published in open access sources) which specifically dealt with camels and their products. 

The number of different journals in which these articles were published was 203 (49 journals or book chapters non-indexed in JCR in the year of publication of their reviewed camel-related papers and 154 JCR indexed journals). Considering the country of the corresponding author for each article, camel research is present in 56 countries around the world.

Regarding specific protection-and-care laws, the World Organization for Animal Health (WOAH) ‘Terrestrial Code’ [19] contemplates camels as livestock and thus has provided concrete welfare recommendations since its first edition in 1968 until the present. Simultaneously, national mandatory regulations that categorically include camels as goods-production livestock were solely endorsed in four countries through the following specific legislations; ‘Decree on Animal Transport’ [20], ‘Animal Welfare Regulations’ [21], ‘Regional Decree on Dromedary Transport Activities’ [22] and ‘Animal Welfare Act’ [23]. No further legislation was found at a national or international level.

### 3.2. Conditioning Factors Analysis

Publications dealing with camels have markedly increased since the beginning of the present century, as shown in Figure 1. The journals in which papers dealing with camels were more frequently represented are summarized in Figure 2.

The highest JCR impact index (23.083) was reached by an article focused on functional heavy-chain antibodies in Camelidae and published in ‘Advances in Immunology’ in 2001 (https://doi.org/10.1016/S0065-2776(01)79006-2). On the other hand, the article with the lowest JCR impact index (0.128), which is about camel-related human injuries, was published in ‘Injury’ in 1994 (https://doi.org/10.1016/0020-1383(94)90152-X). Mean JCR impact index per year between 1992 and 2019 is shown in Figure 3.

The most cited publication (779 cites) focused on single domain camel antibodies and was published in ‘Reviews in Molecular Biotechnology’ in 2001 (https://doi.org/10.1016/S1389-0352(01)00021-6). Contrastingly, 83 scientific publications (with independence from the topic or the periods considered in this study) have received no citations until 31 December 2019. Mean number of citations per year is presented in Figure 3.

For JCR impact factor per paper publication year across journals, significant differences (*p* < 0.05, df = 202) were found between most of the pairwise comparisons. For the other three variables (mean JCR impact factor per journal in the whole period, total number of citations of papers and mean number of citations per journal), there were significant differences (*p* < 0.01, df = 202) between the journals focused on Food Science and Technology, Camel Health and Camel Reproduction (median JCR > 1.5) and the remaining journals (median JCR > 1.5). 

Similarly, when the number of authors participating per publication was considered, significant differences (*p* < 0.05, df = 16) were found for JCR impact factor per paper publication year, mean JCR impact factor per journal and the mean number of citations per journal between publications in which 7–12 or up to five authors had been involved; in the case of the latter, these were indeed the most frequent cases as well. No significant differences (*p* > 0.001, df = 16) were found for total number of citations per paper.

When evaluating scientific impact factor across country of corresponding author, significant differences (*p* < 0.05, df = 55) were found between central and north-east European and Asian and North African countries for the four variables considered. Within Asia, significant differences (*p* < 0.001) were found between Middle Eastern countries and other rather eastern countries of this area (China, Japan, Malaysia and Thailand) with scientific publications in camels. Figure 4 depicts a Quantum Geographical Information System (QGIS) map displaying the number of camel research papers per country to illustrate these results. The relative contribution of papers to camel science depending on the topic/s addressed by such papers is reported in Figure 5.

In relation with the different topics addressed by camel research, there were significant differences (*p* < 0.05, df = 17) for the four variables considered between Animal Health (Parasitic and Infectious Diseases), Food Science and Technology and Camel Reproduction in comparison with Livestock Management and Production, Physiology, Adaptative Ecology and Clinical and Biomedical Research. Figure 6 shows the frequency distribution of specific topics in international camel research. The most common topics (more than 90 publication at least), in increasing order of frequency, were Food Science and Technology, Reproduction, Clinical and Biomedical Research, Parasitic Diseases and Infectious Diseases. Figure 7 represents the evolution of the number of publications within the five most popular topics in camel research on a global scale.

The year of publication factor reported significant differences (*p* < 0.005, df = 85, as no impact factor had been registered for any of the journals before 1992) between the years in the early and mid 20th century and the articles published since 1990 (especially those from 2000 to 2019) for all variables considered.

When camel species conditioning effect was examined, no significant differences were reported (*p* = 0.005, df = 11) for any of the variables studied. However, when frequencies were compared, the number of publications for which *Camelus dromedarius* was the species studied was substantially higher than those for *Camelus bactrianus*.

### 3.3. Chronological Evolution of Specific Legislation and Research Advances

Significant differences (*p* < 0.05, df = 4) were found between JCR impact factor per paper and publication year and mean JCR impact factor per journal in the whole period between scientific publications published between 1880–2006 and the remaining four periods considered in this study. When considering the total number of citations to papers and the mean number of citations per journal, there were significant differences (*p* < 0.05, df = 4) between the papers published between 2016–2019 and the papers published between 1880–2015. The parallel evolution of camel research advances and specific law enforcement are presented in Figure 1 and Figure 7 (red chronological line). 

The WOAH ‘Terrestrial Code’ describes and defines general protocols and directives that may be relevant to camel welfare such as standards for animal transport, their slaughtering process for human consumption and some specific facts and concerns applicable to specific diseases. This ‘Terrestrial Code’ is an annual-edition compendium that includes a user guide intended to help competent authorities and other interested parties worldwide to interpret its regulation content and encourages legislative councils to promote legislative adaptations both at a regional or wider international scale when necessary. In this context, national laws are reduced to a brief, technical-based approach for basic physiological needs satisfaction and handling practices during transportation and slaughtering, both with farmed and feral camels.

### 3.4. Regression Coefficients

Table 3 reports standardized regression coefficients for each of the predictors for which a significant effect (*p* < 0.05) was detected using the Kruskal–Wallis H test. These regression coefficients were used to issue the following equation describing the linear relationship between journal impact factor related variables and the predictors of journal, country of corresponding author, topic and year. Table 4 shows the regression equations that were used and a model summary for the regression equations issued for journal impact factor using standardized data.

## 4. Discussion

### 4.1. Potential Conditioning Factors in Camel Science Progress

The progressive mechanization of agricultural labors relegated the camel to the background in tasks to which it had traditionally been used (to achieve an almost exclusive role as a source for food (meat and milk) or leisure (camel races and tourism)). However, camel cultural relevance has hardly been affected in countries where nomadic tribes still exist and whose survival depends on these animals, whose extended use is also frequently linked to an inefficient transport network [24].

This relegation leads to the progressive social appreciation of these multipurpose animals because of their production performance even under extreme climatic conditions [1], the biomedical applications of some of their derived products [25] and their potential for animal-assisted therapeutics [26]. A notable and competitive increase in the interest provided to the functional potentialities of the species have appeared on the scene singularly in the last 30 years, reaching a mean annual growth of 11.02%, with their maximum value (141.67%) reached in 2008. This academic upturn is simultaneous to an increase in worldwide camel census [27,28] and ratified both by the largest peak in the mean impact factor achieved by the research documents dealing with camels occurring in 2001 (Figure 3) and the upward positive trends in camel science impact since then (Appendix A). In spite of these results, when compared to other domestic livestock, publications on camels are marginal from a quantitative point of view, which is largely due to world censuses on camels being smaller, the limited geographical distribution of these animals and to the fact that markets still continue investing little in products derived from camels given the misconception about them being unproductive animals. Therefore, scientific interventions may need to focus on exploring and implementing actions towards the achievement of new production niche opportunities, addressing challenges/constraints in marketing camel derived products and the efficiency and effectiveness of their commercial distribution chain [29,30].

Fortunately, the relatively high number of open access publications have to be considered as it reflects the trends in camel research. Contrastingly, open access costs may be taken into account as these economic barriers could imply scientists may not be able to afford an effective wide communication. In this context, Freire and Nicol [13] proposed self-archiving and open access policies should be promoted and subsidized as a way to encourage the role of developing countries in future scientific challenges.

The first countries to start paying attention to these animals in this regard were those in which a previous tradition forged in camel breeding and production existed, with the benefits derived from the species occurring at both an economic and socio-cultural level [31]. Our results address a relationship between the countries with the highest number of publications and the above-mentioned estimates of global growth of the dromedary and Bactrian camel populations; that is, a higher number of publications from those countries where there is a positive growth of these animal populations [32] and in which traditional camel breeding and production systems are well established (Figure 4). Therefore, given the economic and cultural role of camels in these geographical regions, the specialization of their research groups and desire to publish in journals of high impact factor would be higher. The demographics and institutional prestige of the corresponding author could also bias submitted manuscript outcomes [33]. As our results suggested and which could be somehow expected, eastern Africa and Middle Eastern countries reached the higher mean impact indexes, provided their tradition in the implementation of research advances and hence the scientific impact of the authors involved in the publications of papers.

Closely related to this and despite of the fact that the species studied appeared not to statistically significantly influence research impact in camel science, the number of publications dealing with *Camelus dromedarius* (n = 491) are substantially more numerous than those dealing with *Camelus bactrianus* (n = 40) and *Camelus ferus* (n = 3). This could be explained by the existence of a greater number of animals in the African continent according to published censuses [34] and because the animals in this continent are the fundamental subsistence base of human populations. In this context, development of research projects is necessary to investigate the different functional niches into which dromedaries can be functionally and potentially improved. On the other hand, Bactrian camels, which are mainly found in the Arabian Peninsula, are less numerous and are mostly confined to racing sports, which, compared to dromedaries, would be one of the main functional niches. Nonetheless, the average research impact index is higher for publications covering common aspects of the genus *Camelus*. Such a finding can be ascribed to the fact that those studies specially focused on one of the three camel extant species and are strongly dependent on the data derived from those focused on common issues in the genus *Camelus* and are thus cited at higher rates.

The progressive increase of the dromedary population in Africa is also linked to the sedentarization of previously nomadic populations in this continent and their initiation to camel breeding, the production for their subsistence and the development projects carried out in these countries since the end of the 20th century [35,36]. The establishment of agropastoral systems due to the closure of important migratory routes derived from the privatization of sections of the drylands for large-scale agriculture forced pastoralists to diversify into agriculture and the market economy. For the scientific community, this would allow access to a greater number of dromedaries under human control, a fact that would facilitate its management and the planning of cooperative research studies.

In this context, when establishing international partnerships and preparing papers to be published, the number of authors involved affects the probability of publishing [37] in highly ranked journals [38]. If a study conjoins the efforts of multiple research centers, data is gathered by multiple persons, cross-sectional studies are carried out, statistical analysis are done by different people, and the complexity of the studies in higher, then it is justified to have an increased number of authors [39]. Furthermore, these types of multi-center studies have higher sample sizes and are published in journals acknowledged with higher impact factors [40]. In particular, in consortium-derivative research, the creation of a solid network as specialized as possible is crucial to deal with the requirements of interdisciplinarity and thematic complexity required by certain research fields [41]. Authors of a high professional background belonging to research institutions of recognized prestige or more advanced countries specialized in certain topics usually play important roles in the coordination and direction of international projects and networks. The magnitude of co-authorship and inter-institutional collaboration can be extrapolated to the quality and quantity of the work and collaboration networks established to carry out an investigation, which consequently implies a greater probability of the results obtained being transferred to society [41] and improves visibility in terms of journal importance [42]. Contrastingly, some authors reject the existence of a positive association between the number of authors and the prestige of the journal when measured by its impact factor as this association is evaluated for a specific country or area, which may bias the results [41]. These authors claim that if sample size is reduced, the study does not involve a multi-center background, and having a large number of authors may not be justified.

On the other hand, when evaluating research impact at a minor scale such as the total citations to each published paper, the number of authors does not necessarily condition it. Accordingly to this statement and our particular findings, the above-average visibility achieved by co-authored publications measured in terms of citations received is not an objective criteria when predicting research impact as self-citation can magnify it, especially when research is performed by more authors and from distinct institutions [41].

In summary, our results show the average impact is greater for the publications in which 7–12 authors are involved. Research groups from countries with traditional camel breeding and production systems (Africa and Middle Eastern countries), mainly involved in world-level cooperating projects for its specialty in the subject, reported the highest scientific outcomes. The greatest advances in camel international research were produced in the areas of Food Science and Technology, Camel Health (Infectious and Parasitic Diseases) and Camel Reproduction.

For the particular case of Clinical and Biomedical Research, although scientific papers in this applied field of research are one of the more numerous within camel science, their impact remains low. Although it can be considered as an emerging research topic due to its radical novelty and relatively fast growth, its novelty and the need for *standardization* of related methodologies may be conditioning its impact and dissemination in high-impact multidisciplinary journals that could favor a broader visibility of this promising research field among the scientific community [43,44]. Such findings are also supported by the upturn of the topic ‘Food Science and Technology’ over the ‘Clinical and Biomedical’ discipline since the early 20th century (Figure 6), as prior to biomedical research for camel-derived products, it may have been useful and preferable to perform a characterization of these products, of which Food Science and Technology may be responsible.

Meanwhile, topics such as Management, Nutrition, Ecology, Genetic Management and Production are scarcely approached [45] and likewise, the results of these research topics are often inconclusive due to reduced sampling, scant observations or weak statistical treatment of data [5]. This situation can be conditioning the current limited interest of the scientific community in camel welfare because of the close relationship between animal wellbeing and food-producing systems in the emerging scenario of this industry. Consequently, production-related topics must be urgently reoriented and assessed due to production–health–welfare complex interactions.

### 4.2. Camel Research Advances’ Impact on Extended Related Legislation

Until the late 1990s, camel production was mainly based on a traditional husbandry system ascribed to rural livelihoods whose main purpose was the obtention of derived products such as meat and milk. Although the notorious impact of camel research, especially in the last two decades, clearly underlines the increase in the economic interests in camel breeding and the progressive technification of rearing systems for their productive potentialities [46], scarce scientific attention has been paid to such related topics (production and behavior physiology) (Figure 7). In the absence of quantitative information regarding the level of camel welfare in different housing systems, their basic needs may not be fully satisfied and their productive potential and profitability are devalued as their general health status might be affected by neglected practices. In this context, research impact factor related variables must be considered carefully when we aim to determine their potential conditioning effect on the evolution of regulations. Studies have shown that research documents need at least two to three years after publication to be cited enough for bibliometric indicators to be reliable and citations also continue accumulating over time [47,48]. This means that older papers are more cited than younger ones just because they have had more time to accumulate citations, but not necessarily for their scientific impact or improvement they may entail.

As animal welfare could comprise and be conditioned by multiple factors such as different people’s view or animal scientists’ perspectives, it becomes crucial to present and identify common and opposing points of view to achieve the most comprehensive worldwide-accepted definition. Participative dialogues for extensive discussions involving industry, welfare research groups, experienced advisers and lawmakers have to be proposed so as to approach farmed camels’ welfare and address the most appropriate ways to maximize their efficiency and productivity in a sustainable manner. For this purpose, advances in the understanding of animal physiology and behavior, technological changes in animal husbandry and their relationship to the welfare of animals must be taken into account [49].

Since derived legislation in poultry, pigs, cattle, sheep, goats and fur animals’ welfare is widely available, specific regulations and guidance concerning the welfare of reared camels are noticeably limited [50]. At an international level, the World Organisation for Animal Health (OIE) ‘Terrestrial Code’, an annual-edition compendium, describes and defines general, brief protocols and directives that may be relevant to camel welfare such as standards for animal transport, their slaughtering process for human consumption and some specific facts and concerns applicable to specific diseases. However, this ‘Terrestrial Code’ includes specific chapters containing detailed minimum requirements and recommendations for cattle, chicken, equid and pig welfare depending on production systems or regimes in which reared, from birth through to finishing; that is, specifically animal-based criteria or measurables that can be useful indicators of animal welfare and other outcome-based recommendations (i.e., biosecurity, environmental conditions and management practices). For camels, this information is lacking. Such a finding provides insight into the need for camel science to be reinforced in closely related topics, as the OIE standards are based on the most recent scientific articles in light of advances in veterinary science. Camels are only considered in this global code for transportation and slaughtering purposes.

When seeking both animal and human co-existence and prosperity, nationally competent authorities should promote alternatives and implement research and development projects for existing animals’ sustainable exploitation. Only four countries in the world have enforced internal compulsory regulations on minimum, shallow requirements for the farming, transportation and slaughtering of camels. Both farmed and feral social awareness is presumably very low, which ratifies that certain camel science approaches remain little-known. In addition, global animal welfare councils are not encouraging national lawmakers to undertake specific mandatory regulations. Therefore, camel keepers may not be overcoming challenges provided by their emerging condition and the lack of specific legislation for these animals in terms of animal wellbeing.

Under this framework, while innovative legislation is being drawn up, well-organized camel industries are expected to demand high voluntary provisional welfare standards from their human resources and research needs to be compulsorily considered in the process. In this sense, the scientific community plays an additional role to prevent livestock producers from starting to think like business people for whom sustainable and good husbandry practices could be disregarded as a result of animals being considered mere economic products. By recognizing the positive effects of animal wellbeing on production rates, the public’s perception of the livestock industry as a whole will improve, and the resultant regulations of collaborative conventions will be extended.

Countries where camel censuses are significantly higher and/or their outstanding research potential is widely recognized could be suitable for promoting large research consortia on a global scale. These consortia will be formed by solid entities to play advisory roles in camel welfare science using their direct empirical experience derived from the analysis of large samples, which may maximize the validity of their conclusive results. In turn, this may translate into the potential influence and interpretation of camel literature for policy purposes to promote the access to the financial resources for academics to carry out their research.

## 5. Conclusions

Despite animal welfare scientific interests having grown considerably due to consumers’ concern worldwide, it remains overlooked in some minor species, such as farmed camels. Maintaining ethically acceptable conditions in these animals when reared requires the establishment of evidence-based guidelines measuring environmental and animal-based welfare indicators and scores. These implementations may lead to the prosperity of the species and its relationship with humans through the achievement of sufficient camel welfare outcomes. The present research highlights the current request for in-depth and constructive intercommunication between camel breeders, consumers, scientists and policymakers. This increase in communication must be implemented to seek the commitment for law enhancement to address specific emerging needs in these multipurpose animals. This situation may make camel production and functionality based on socially, economically sustainable production systems on a global scale. In this context, the relationship between science and authorities should become the leitmotif on which to rely to face and approach not only the productive opportunities, but also priority challenges and sustainable improvements to ensure the long-term future of camels. The world’s highest-ranked research institutions in camel science are cardinal when establishing this *pluridirectional communication* interface due to the high *research* performance that they present, their innovation outputs and their impact in society.

## Figures and Tables

**Figure 1 animals-10-00780-f001:**
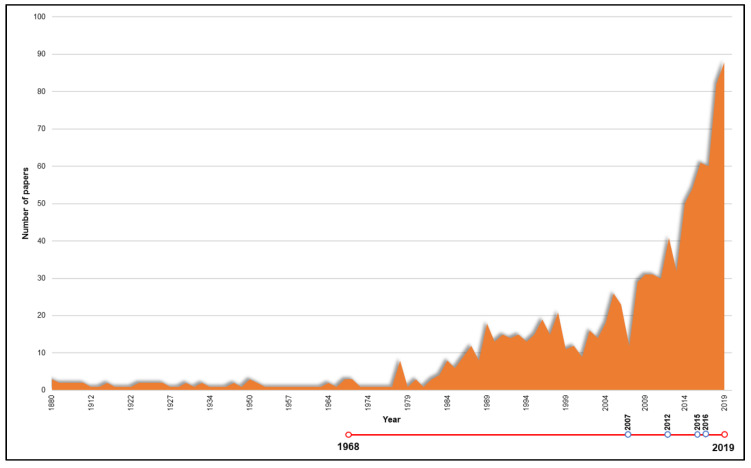
Number of camel research publications from 1880 to 2019. Timeline is represented below the graph in red, with blue-contoured spots marking the moment of release of a regulation document.

**Figure 2 animals-10-00780-f002:**
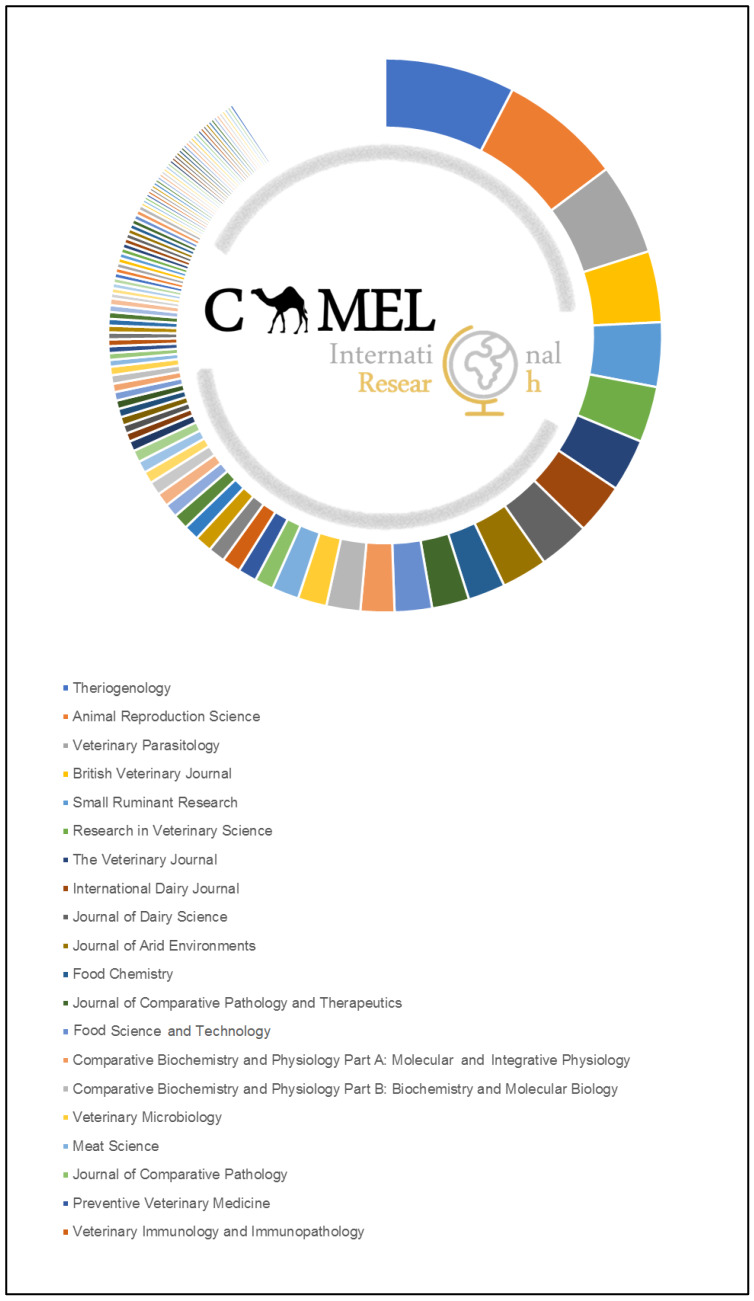
Number of historical publications on camels across journals.

**Figure 3 animals-10-00780-f003:**
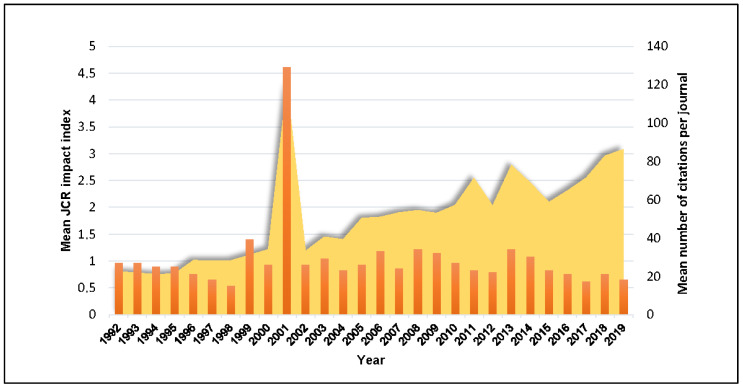
Mean JCR impact index and mean number of citations per year between 1992 and 2019.

**Figure 4 animals-10-00780-f004:**
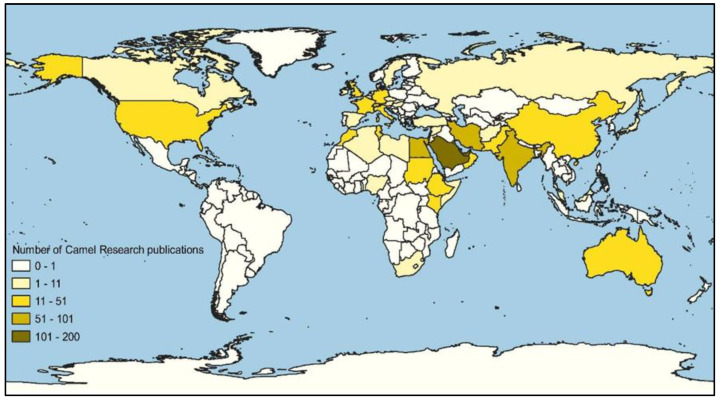
Quantum Geographical Information System (QGIS) map displaying the number of camel research papers per country.

**Figure 5 animals-10-00780-f005:**
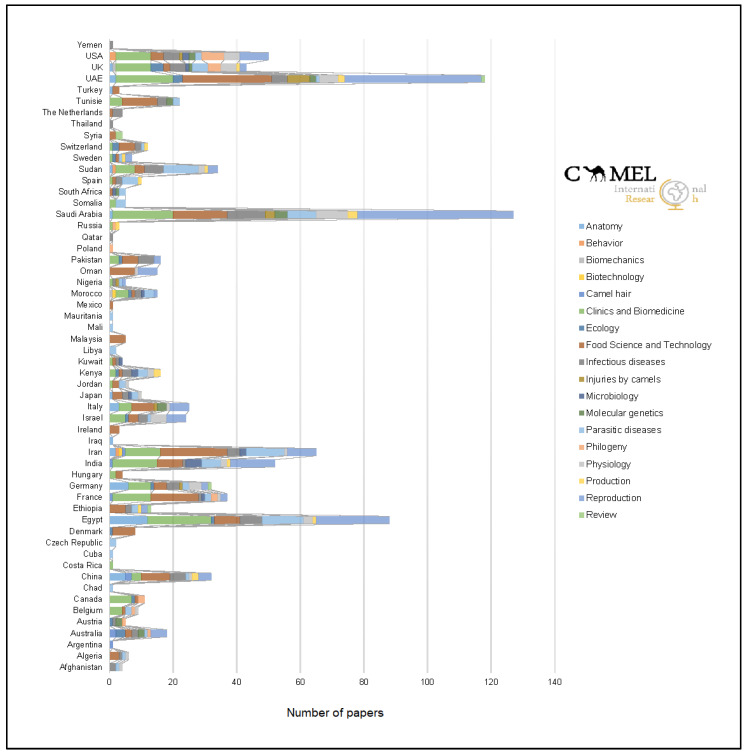
Number of scientific publications per country and research item within camel science.

**Figure 6 animals-10-00780-f006:**
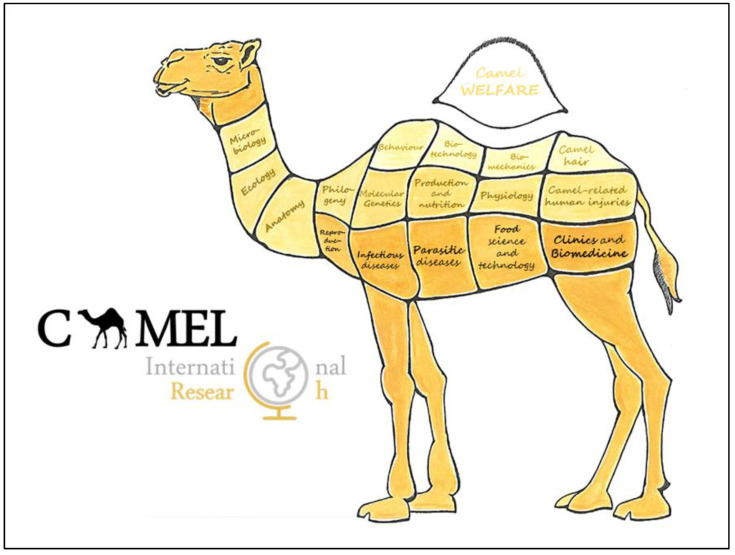
Specific topics in camel international research. Color intensity is relative to the number of publications in each scientific area. The darker the color, the higher the number of publications dealing with that topic.

**Figure 7 animals-10-00780-f007:**
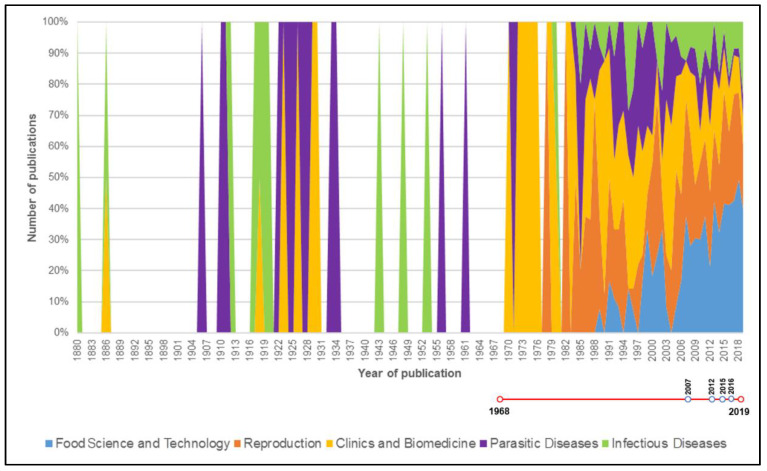
Numerical evolution of the five most popular items in international camel research from 1880 to 2019. Topics with publication numbers below 90 were omitted to improve the interpretability of the results. Timeline is represented below the graph in red, with blue-contoured spots marking the moment of release of a regulation document.

**Table 1 animals-10-00780-t001:** Category description for conditioning factors considered to classify camel bibliography.

Factor	Type	Levels
Journal	Nominal	203 Scientific Journals
Number of Authors	Ordinal	From 1 to 15, 20 and 24
Country of Corresponding Author	Nominal	56 countries
Topic	Nominal	Anatomy, Behaviour, Biomechanics, Biotechnology, Camel Hair, Clinical and Biomedical Research, Ecology, Food Science and Technology, Infectious Diseases, Camel-Related Human Injuries, Microbiology, Molecular Genetics, Parasitic Diseases, Phylogeny, Physiology, Production, Reproduction and General Review
Year of Publication	Ordinal	1880 to 2019
Camel Species or Species Cluster	Nominal	*Camelops* sp. (extinct) ^1^, *Camelops hesternus* (extinct) ^1^, *Camelus bactrianus*, *Camelus dromedarius*, *Camelus ferus*, *Camelus knoblochi* (extinct) ^1^, *Camelus* sp., *Camelus thomasi* (extinct) ^1^, *Megatylopus* sp. (extinct)1, *Paracamelus aguirrei* (extinct)^1^, Species cluster 1 (*Paracamelus* sp. (extinct) ^1^, *C. hesternus* (extinct) ^1^, *C. bactrianus*, *C. dromedarius*) and Species cluster 2 (*C. bactrianus*, *C. dromedarius*)

^1^ Study performed on fossils or stuffed collections.

**Table 2 animals-10-00780-t002:** Mean, mode and interquartile range (IQR) for the variables of Journal Citation Report (JCR) impact factor per paper publication year, mean JCR impact factor per journal during the whole period, total citations of the paper, mean number of citations per journal and predictors in the model designed.

Variable	Median	Mode	IQR
JCR impact factor per paper publication year	1.50	0.00	2.30
Mean JCR impact factor per journal in the whole period	1.32	0.00	1.42
Total citations of the papers	12.00	0.00	25.00
Mean number of citations per journal	20.00	20.00	16.56
Journal	102	181	127
Number of authors	4	3	4
Country of corresponding author	30	41	30
Area/Topic	9	8	7
Year of publication	2010	2019	19
Camel species	4	4	4

**Table 3 animals-10-00780-t003:** Standardized regression coefficients for each of the predictors reporting a significant value (*p* < 0.05) by Kruskal–Wallis H test.

JCR Impact Factor per Paper Publication Year	β	0.632 Bootstrap Estimate of Std. Error	df	F	Significance
Journal	0.732	0.028	202	660.910	0.001
Number of Authors	0.089	0.032	3	7.785	0.001
Country of Corresponding Author	0.156	0.026	55	36.065	0.001
Area/Topic	0.266	0.064	17	17.260	0.001
Year of Publication	0.575	0.030	17	359.966	0.001
$Y_{\mathrm{JCR}/\mathrm{journal}/\mathrm{year}}=0.732\cdot X_{\mathrm{Journal}}+0.089\cdot X_{\mathrm{Authornumber}}+0.156\cdot X_{\mathrm{Country}}+0.266\cdot X_{\mathrm{Topic}}+0.575\cdot X_{\mathrm{Year}}$, where X is the observation for each of significant predictors encoded as a number.					
**Mean JCR Impact per Journal in the Whole Period**	**β**	**0.632 Bootstrap Estimate of Std. Error**	**df**	**F**	**Significance**
Journal	0.977	0.009	202	11,022.598	0.001
Number of Authors	0.001	0.055	16	0.001	1.000
Country of Corresponding Author	0.058	0.032	55	3.288	0.001
Area/Topic	0.100	0.036	17	7.457	0.001
Year of Publication	0.155	0.033	82	21.577	0.001
$Y_{\mathrm{JCR}/\mathrm{journal}}=0.977\cdot X_{\mathrm{Journal}}+0.058\cdot X_{\mathrm{Country}}+0.100\cdot X_{\mathrm{Topic}}+0.155\cdot X_{\mathrm{Year}}$, where X is the observation for each of significant predictors encoded as a number.					
**Total Citations of the Paper**	**β**	**0.632 Bootstrap Estimate of Std. Error**	**df**	**F**	**Significance**
Journal	0.735	0.099	192	550.010	0.001
Number of Authors	0.342	0.166	15	40.252	0.001
Country of Corresponding author	0.290	0.073	53	150.724	0.001
Area/Topic	0.397	0.133	16	80.972	0.001
Year of Publication	0.403	0.080	80	250.362	0.001
$Y_{\mathrm{totalcitations}}=0.735\cdot X_{\mathrm{Journal}}+0.342\cdot X_{\mathrm{Authornumber}}+0.290\cdot X_{\mathrm{Country}}+0.397\cdot X_{\mathrm{Topic}}+0.403\cdot X_{\mathrm{Year}}$, where X is the observation for each of significant predictors encoded as a number.					
**Mean Number of Citations per Journal**	**β**	**0.632 Bootstrap Estimate of Std. Error**	**df**	**F**	**Significance**
Journal	0.987	0.007	202	19,778.36	0.001
Number of Authors	0.001	0.039	4	0	1.000
Country of Corresponding Author	0.065	0.031	55	4.465	0.001
Area/Topic	0.062	0.031	17	3.988	0.001
Year of Publication	0.067	0.02	82	11.186	0.001
$Y_{\mathrm{meancitations}/\mathrm{journal}}=0.987\cdot X_{\mathrm{Journal}}+0.065\cdot X_{\mathrm{Country}}+0.062\cdot X_{\mathrm{Topic}}+0.067\cdot X_{\mathrm{Year}}$, where X is the observation for each of significant predictors encoded as a number.					

**Table 4 animals-10-00780-t004:** Model Summary for Categorical Regression of journal impact factor using standardized data.

**JCR Impact Factor per Paper Publication Year**		**Multiple R**	**R^2^**	**Adjusted R^2^**
0.952	0.906	0.852
**ANOVA**	**Sum of Squares**	**df**	**Mean Square**	**F**	**Significance**
Regression	916.288	372	2.463	16.592	0.001
Residual	94.712	638	0.148		
Total	1011	1010			
**Mean JCR Impact per Hournal in the Whole Period**		**Multiple R**	**R^2^**	**Adjusted R^2^**
0.999	0.999	0.999
**ANOVA**	**Sum of squares**	**df**	**Mean square**	**F**	**Significance**
Regression	1010.972	372	2.718	62,822.211	0.001
Residual	0.028	638	0.000		
Total	1011	1010			
**Total Citations of the Paper**		**Multiple R**	**R^2^**	**Adjusted R^2^**
0.901	0.812	0.695
**ANOVA**	**Sum of squares**	**df**	**Mean square**	**F**	**Significance**
Regression	753.584	356	2.117	6.930	0.001
Residual	174.416	571	0.305		
Total	928.000	927			
**Mean Number of Citations per Journal**		**Multiple R**	**R^2^**	**Adjusted R^2^**
0.999	0.999	0.999
**ANOVA**	**Sum of squares**	**df**	**Mean square**	**F**	**Significance**
Regression	1009.944	372	2.715	30,624.972	0.001
Residual	0.056	637	0

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
