# Peer review of "Effect of Research Impact on Emerging Camel Husbandry, Welfare and Social-Related Awareness"

_animals, 2020, doi:10.3390/ani10050780_

Round 1

Reviewer 1 Report

Please find comments in the attached file.

Author Response

Open Review

English language and style

( ) Extensive editing of English language and style required
( ) Moderate English changes required
(x) English language and style are fine/minor spell check required
( ) I don't feel qualified to judge about the English language and style

Yes

Can be improved

Must be improved

Not applicable

Does the introduction provide sufficient background and include all relevant references?

(x)

( )

( )

( )

Is the research design appropriate?

(x)

( )

( )

( )

Are the methods adequately described?

(x)

( )

( )

( )

Are the results clearly presented?

(x)

( )

( )

( )

Are the conclusions supported by the results?

( )

(x)

( )

( )

Comments and Suggestions for Authors

Please find comments in the attached file.

Reviewer comments on the manuscript ID animals-784820 entitled ‘Effect of research impact on

emerging camel husbandry, welfare and social-related awareness’

General comments

A much-improved version of the manuscript. I appreciate the efforts put in by the authors to

improve the manuscript as per the review comments.

I still find too long and verbose sentences. However, leave it to the discretion of the editor to Accept them or not.

I recommend the publication of the manuscript after incorporating the suggestions/ comments.

Response: We thank the reviewer for his/her kind comments.

Specific comments

Introduction

Lines 24-25: Please reframe ‘…. renders impractical the efforts to ensure sustainable camel

husbandry practices under the scope of welfare’ as ‘renders the efforts to ensure sustainable camel husbandry practices under the scope of welfare impractical.

Response: Reframed.

Lines 27-29: It is a too long sentence. Please reframe ‘The lack of applied scientific research on

camels despite they being recognized as production animals compels the reorganization of emerging camel breeding systems aiming to achieve successful camel welfare management strategies all over the world’ as ‘The lack of applied scientific research on camels despite they being recognized as production animals compels the reorganization of emerging camel breeding systems. These aim to achieve successful camel welfare management strategies all over the world’

Response: Reframed.

Lines 38-39: I am not able to understand these lines ‘…so intentional harming acts and less clear-cut situations neglecting basic needs may persist in these species’ Please clarify or reword.

Response: Sentence was reworded.

Introduction

Line 51: Please delete ‘in’.

Response: Deleted.

Lines 71-73: Please reword this sentence. I suspect there is a missing word in the indicated space. ‘Besides, while nomadic pastoralists are challenging sedentarization, backward nomad livestockbreeding does …………. into socialist livestock-breeding within intensive inputs for different purposes’

Response: Reworded.

Line 82: Please delete ‘for’.

Response: Deleted.

Lines 88-90: Please reframe ‘Camel intensive management could be affecting their natural

behavioral repertoire and making camels develop stereotypies’ as ‘Intensive management could be affecting their natural behavioral repertoire leading to the development of stereotypies’

Response: Reframed.

Line 96: Please replace ‘stand out’ with ‘identify’

Response: Replaced.

Line 106: Please delete ‘fastly’

Response: Deleted.

Lines 105-07: Please reword’ Bibliometric mapping enables to fastly visualize scientific developments and which involved parties might play important roles in the different subfields’ as ‘Bibliometric mapping enables visualization of scientific developments leading to an active involvement of stakeholders in the different subfields’

Response: Reworded.

Line 107: Please replace ‘Second’ with’ Secondly’

Response: Replaced.

Line 114: Please delete ‘also appeal to and’

Response: Deleted.

Materials and methods

Lines 149-50: I am not able to understand this part of the sentence ‘..and strength both advice and responsibility for persons or leaders in charge of them’. Please clarify.

Response: Clarified.

Line 170: Please replace ‘First’ with ‘Initially’

Response: Changed.

Results

Line 331: Please check spelling ‘Cronological’ and elsewhere in the manuscript.

Response: Corrected across manuscript.

Lines 339-47: I feel these are not relevant to the results section. Please delete and refer to this code in the discussion.

Response: Reviewer’s suggestion was followed.

Discussion

Line 367: Please replace ‘yields’ with ‘performance’

Response: Replaced.

Line 380: Please reword ‘productive niches opportunities’ as ‘production niches’

Response: Reworded.

Line 381: Please reword ‘in camel derived products’ as ‘in marketing camel derived products’

Response: Reworded.

Line 401: Please replace ‘prestige’ with ‘scientific impact’

Response: Replaced.

Lines451-57: It is a huge sentence. Please reframe it into smaller sentences to make it more

comprehensible.

Response: Reframed.

Line 499: Please replace ’take’ with ‘taken’

Response: Replaced.

Line 512: Please replace ‘national’ with ‘nationally’

Response: Replaced.

Line 516: Please reword ‘Social awareness is presumably not as desired’ as ‘Social awareness is

presumably very low’

Response: Reworded.

Submission Date

10 April 2020

Date of this review

24 Apr 2020 05:07:29

Reviewer 2 Report

I would like to thank the authors for incorporating the suggestions provided. I think the quality of the manuscript has improved overall.

Some minor suggestions are:

Line 126 please add retrospective observational longitudinal study

Line 156 please clarify if the Shapiro Wilk or the Shapiro-Francia test was applied.

L173. The total number of citations of each paper??

Through out the text replace cronological with chronological

Replace WOAH with OIE, although the organization changed its name they kept  the acronym OIE.

The text seems much more organized and easier to follow, the figures are also more informative now.

Author Response

Review Report Form

Open Review

English language and style

( ) Extensive editing of English language and style required
( ) Moderate English changes required
( ) English language and style are fine/minor spell check required
(x) I don't feel qualified to judge about the English language and style

Yes

Can be improved

Must be improved

Not applicable

Does the introduction provide sufficient background and include all relevant references?

(x)

( )

( )

( )

Is the research design appropriate?

( )

(x)

( )

( )

Are the methods adequately described?

(x)

( )

( )

( )

Are the results clearly presented?

(x)

( )

( )

( )

Are the conclusions supported by the results?

(x)

( )

( )

( )

Comments and Suggestions for Authors

I would like to thank the authors for incorporating the suggestions provided. I think the quality of the manuscript has improved overall.

Response: We thank the reviewer for his/her kind comments.

Some minor suggestions are:

Line 126 please add retrospective observational longitudinal study

Response: Added.

Line 156 please clarify if the Shapiro Wilk or the Shapiro-Francia test was applied.

Response: Clarified.

L173. The total number of citations of each paper??

Response: Clarified.

Through out the text replace cronological with chronological

Response: Replaced.

Replace WOAH with OIE, although the organization changed its name they kept  the acronym OIE.

Response: Replaced.

The text seems much more organized and easier to follow, the figures are also more informative now.

 Response: Thank you.